# Imaging Findings of Clinical Significance in Endodontics During Cone Beam Computed Tomography Scanning of the Upper Airway—The Anterior, Bilateral, C-Shaped, Dual of Mandibular Root Canals: A Brief Case Report

**DOI:** 10.3390/diagnostics15243157

**Published:** 2025-12-11

**Authors:** Edgar García-Torres, Diana Laura Grissel Guerrero-Falcón, Hugo Alejandro Bojórquez-Armenta, Oscar Eduardo Almeda-Ojeda, Víctor Hiram Barajas-Pérez, Luis Javier Solís-Martínez

**Affiliations:** 1Research Group, UJED-CA-130, Faculty of Dentistry, Universidad Juárez del Estado de Durango (UJED), Durango 34070, Mexico; oscar.almeda@ujed.mx (O.E.A.-O.); hiram.barajas@ujed.mx (V.H.B.-P.); javier.solis@ujed.mx (L.J.S.-M.); 2Endodontics Specialty, Faculty of Dentistry, Universidad Juárez del Estado de Durango (UJED), Durango 34070, Mexico; dianagro1197@gmail.com; 3Department of Endodontics, Faculty of Dentistry, Universidad Autónoma de Sinaloa (UAS), Culiacán 81254, Mexico; endobojorquez@gmail.com

**Keywords:** cone-beam computed tomography, airway, root canal, anatomy, incidental findings, bilateral, C-shape, dual, mandible, CARE guidelines

## Abstract

Cone beam computed tomography (CBCT) is a valuable diagnostic tool for evaluating the upper airway and maxillofacial region. This report demonstrates the clinical value of CBCT in identifying significant anatomical variations in endodontics, incidentally detected on a non-endodontic CBCT scan. A 23-year-old female patient underwent CBCT imaging at the Faculty of Dentistry-UJED to evaluate her upper airway. CBCT imaging revealed a unique, complex, and unusual anatomy of mandibular root canals, characterized by Vertucci’s type III root canals in the anterior sextant and co-occurrence of bilateral C-shaped mandibular second molars (type C2 according to Fan’s classification). No therapeutic interventions were initiated due to the patient’s asymptomatic status. CBCT imaging is a valuable tool for integrated diagnostic approaches, underscoring its role in thorough patient management. The integration of multidisciplinary interpretation of CBCT data can enhance diagnostic accuracy and optimize patient records and management, emphasizing the importance of collaborative efforts between radiologists, clinicians, and endodontists. Documenting and sharing such findings can increase awareness of rare anatomical variations, facilitating detection and contributing to medical knowledge.

**Figure 1 diagnostics-15-03157-f001:**
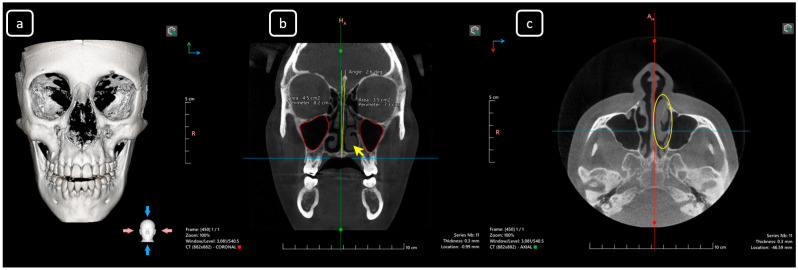
CBCT data of a 23-year-old healthy female from Durango, Mexico, with no known personal or hereditary diseases. The scan was performed for upper airway evaluation as requested by her physician; the patient was asymptomatic for pain or symptoms related to the area. The data includes 3D volumetric reconstruction and multiplanar views. (**a**) A 3D skull reconstruction depicting maxillofacial anatomy. (**b**) Coronal section: The coronal plane reveals the upper airway and maxillary sinus morphology, showing a nasal septum deviation of 2.6° to the left (yellow lines) relative to the midline (green line), with the palatal midline and *crista galli* serving as reference points. The left maxillary sinus has a reduced area of 3.5 cm^2^ compared to the contralateral sinus (red line shapes). Marked hypertrophy of the left inferior turbinate is also observed (yellow arrow). (**c**) Axial section: The axial plane shows the extent of the left inferior turbinate hypertrophy (circled in yellow) and its anteroposterior extension.

**Figure 2 diagnostics-15-03157-f002:**
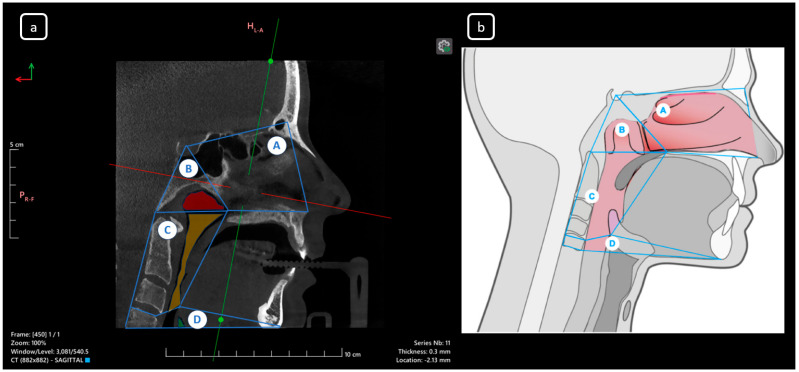
Multiplanar view of the upper airway morphology: (**a**) sagittal CBCT section and (**b**) schematic illustration of corresponding anatomical regions. Labels: (A) Nasal cavity, (B) Nasopharynx, (C) Oropharynx, (D) Hypopharynx [1]. The assessment of upper airway anomalies is crucial for diagnosis, prognosis, and treatment planning, typically performed by specialists. In our institution, the Faculty of Dentistry-UJED, general examinations and structure identification are conducted in the Diagnostic Imaging Department for educational purposes. Additionally, these examinations are integral components of comprehensive patient assessments and postgraduate training in endodontic specialty, ultimately facilitating accurate diagnoses and informing effective treatment planning for optimal patient outcomes. Nasal Airway Obstruction (NAO) is a prevalent condition requiring evaluation, including physical examination and symptom assessment. Septal deviation (76%) and inferior turbinate hypertrophy (72%) are common findings [2]. In the context of diagnostic imaging, this report exemplifies the diagnostic utility of CBCT evaluation in identifying anatomical variations within the maxillofacial region. Additionally, subsequent examination revealed unusual root canal morphologies in the mandibular teeth of significant clinical relevance in endodontics, emphasizing the imperative of comprehensive CBCT imaging analysis. Such variability is exemplified by the A-B-C-D concept, which embodies a complex paradigm of anatomical variability, where multiple, seemingly disparate anomalies converge to create a singular clinical entity. This intricate constellation of variations is characterized by the following key features: Anterior atypical anatomy (A): Mandibular incisors and canines may display unusual root canal configurations, including supplementary canals, aberrant trajectories, or unexpected apical terminations. Bilateral symmetry (B): Identical anomalies are mirrored on contralateral sides of the dental arch, which can be related to genetic or environmental influences on morphogenesis. C-shaped morphology (C): Mandibular second molars display a complex, C-shaped canal system, wrapping around the distal root and creating a formidable challenge for chemomechanical debridement and obturation. Dual nature of root canal systems (D): The “1-2-1” configuration exemplifies the intricate variability of root canal anatomy, featuring a single canal that bifurcates and reunites, creating a complex network of channels, anastomoses, and apical deltas. This A-B-C-D constellation of anatomical variations demands a high degree of diagnostic acumen, technical expertise, and adaptability from the clinician, as well as a profound appreciation for the intricate complexity and variability of the human dentition. The co-occurrence of these variations is exceedingly rare, emphasizing the need for thorough CBCT evaluation and clinician awareness in identifying such complex anatomical presentations. These variations defy conventional norms and challenge traditional endodontic dogma.

**Figure 3 diagnostics-15-03157-f003:**
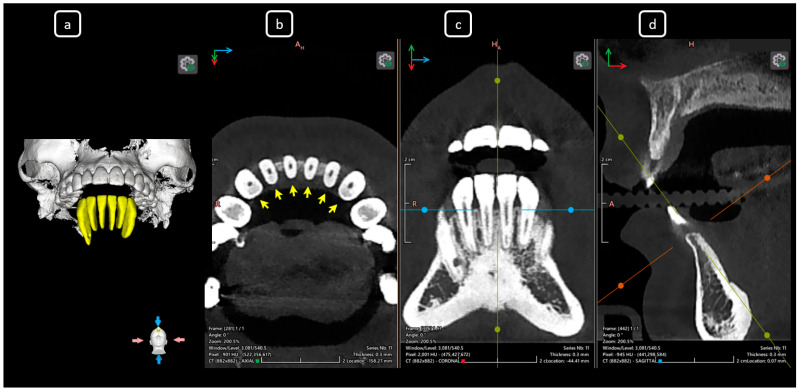
CBCT data with volumetric reconstruction and multiplanar views. (**a**) A 3D reconstruction of the mandibular anterior sextant (in yellow). (**b**) Axial section at the cervical level, showing the entrance of root canals (yellow arrows). (**c**) Coronal section. (**d**) Sagittal section. Note that CBCT sections are not aligned with the individual longitudinal axes of teeth.

**Figure 4 diagnostics-15-03157-f004:**
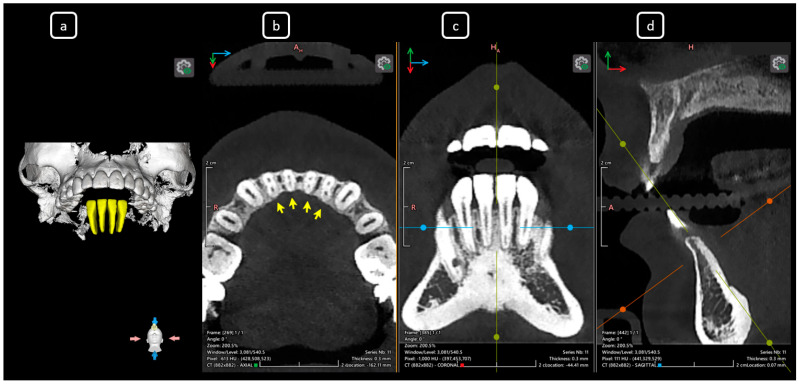
CBCT data with volumetric reconstruction and multiplanar views. (**a**) A 3D reconstruction of teeth 3.2 to 4.2 [3] (in yellow). (**b**) Axial section at the mid-third radicular level of the mandibular incisors, showing the division of the main root canal and the duality of canals in a single root, yellow arrows. The [A], [B], and [D] of mandibular root canals. (**c**) Coronal section. (**d**) Sagittal section.

**Figure 5 diagnostics-15-03157-f005:**
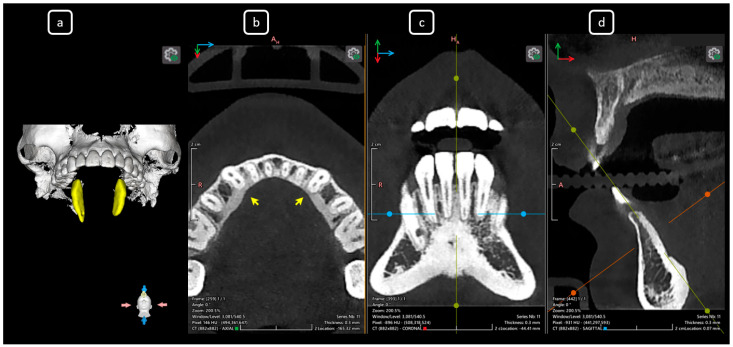
CBCT data with volumetric reconstruction and multiplanar views. (**a**) A 3D reconstruction of teeth 3.3 and 4.3 [3] (in yellow). (**b**) Axial section at the mid-third radicular level of the mandibular canines, revealing bilateral symmetry of dual root canals originating from a single main canal within a single root, yellow arrows. The [A], [B], and [D] of mandibular root canals. (**c**) Coronal section. (**d**) Sagittal section.

**Figure 6 diagnostics-15-03157-f006:**
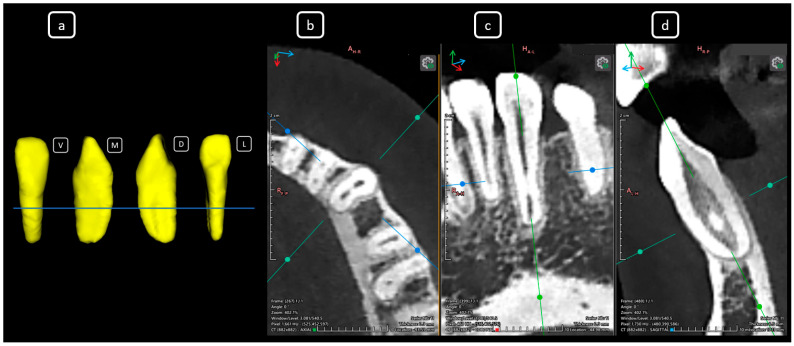
CBCT data with volumetric reconstruction and multiplanar views demonstrate clinically relevant endodontic findings in the mandibular anterior root canals. (**a**) A 3D reconstruction of tooth 3.3 [3] (V: vestibular, M: mesial, D: distal, L: lingual); the blue line indicates the axial slice level. (**b**) Axial section. (**c**) Coronal section. (**d**) Sagittal CBCT section showing Vertucci’s type III [4] canal configuration (1-2-1), characterized by a single root with a main canal that divides and then merges into a single canal apically. Sections are oriented along the tooth’s longitudinal axis (coronal, sagittal).

**Figure 7 diagnostics-15-03157-f007:**
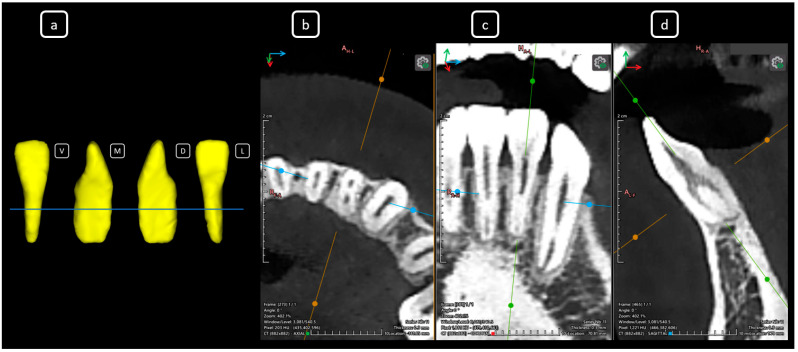
CBCT data with volumetric reconstruction and multiplanar views reveal clinically relevant endodontic findings in the mandibular anterior root canals. (**a**) A 3D reconstruction of tooth 3.2 [3] (V: vestibular, M: mesial, D: distal, L: lingual); the blue line indicates the axial slice level. (**b**) Axial section. (**c**) Coronal section. (**d**) Sagittal CBCT section showing Vertucci’s type III [4].

**Figure 8 diagnostics-15-03157-f008:**
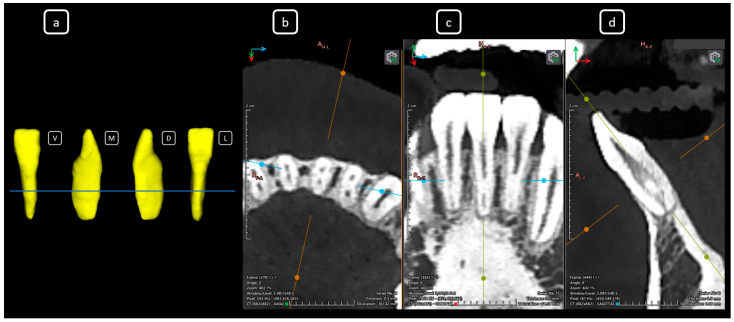
CBCT data with volumetric reconstruction and multiplanar views showcasing clinically relevant endodontic findings in the mandibular anterior root canals. (**a**) A 3D reconstruction of tooth 3.1 [3] (V: vestibular, M: mesial, D: distal, L: lingual); the blue line indicates the axial slice level. (**b**) Axial section. (**c**) Coronal section. (**d**) Sagittal CBCT section showing Vertucci’s type III [4].

**Figure 9 diagnostics-15-03157-f009:**
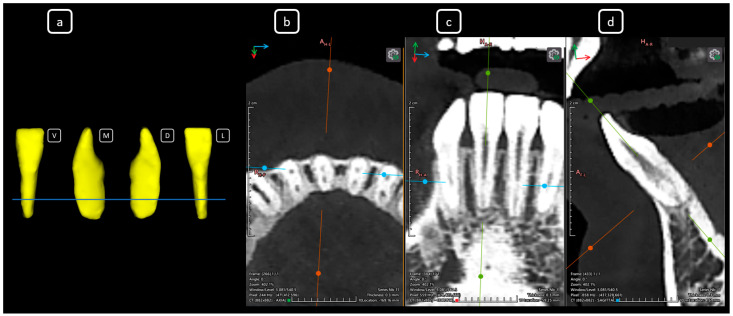
CBCT data with volumetric reconstruction and multiplanar views highlighting clinically relevant endodontic findings in the mandibular anterior root canals. (**a**) A 3D reconstruction of tooth 4.1 [3] (V: vestibular, M: mesial, D: distal, L: lingual); the blue line indicates the axial slice level. (**b**) Axial section. (**c**) Coronal section. (**d**) Sagittal CBCT section showing Vertucci’s type III [4].

**Figure 10 diagnostics-15-03157-f010:**
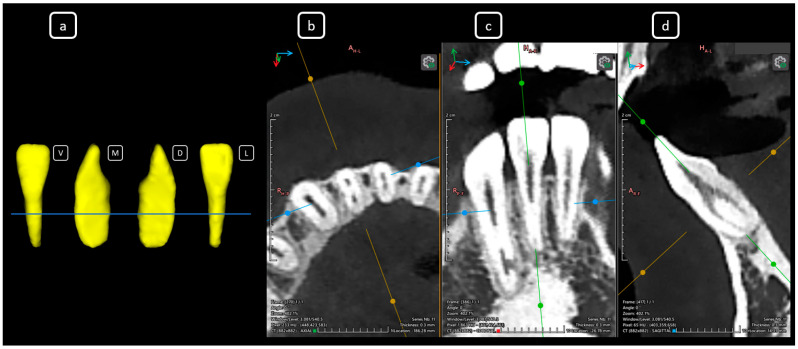
CBCT data with volumetric reconstruction and multiplanar views featuring clinically relevant endodontic findings in the mandibular anterior root canals. (**a**) A 3D reconstruction of tooth 4.2 [3] (V: vestibular, M: mesial, D: distal, L: lingual); the blue line indicates the axial slice level. (**b**) Axial section. (**c**) Coronal section. (**d**) Sagittal CBCT section showing Vertucci’s type III [4].

**Figure 11 diagnostics-15-03157-f011:**
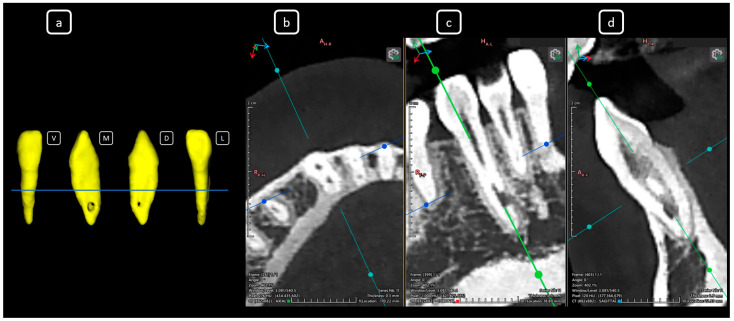
CBCT data with volumetric reconstruction and multiplanar views presenting clinically relevant endodontic findings in the mandibular anterior root canals. (**a**) A 3D reconstruction of tooth 4.3 [3] (V: vestibular, M: mesial, D: distal, L: lingual); the blue line indicates the axial slice level. (**b**) Axial section. (**c**) Coronal section. (**d**) Sagittal CBCT section showing Vertucci’s type III [4]. In addition to the unusual root -canal anatomy, an incidental finding of a hypodense area in the apical portion (c and d) is noted, and a surface defect (a) indicative of root resorption associated with tooth 4.3 is confirmed (Figure 11). Apical root resorption, whether internal or external, frequently exhibits a clinically silent progression in its early stages, often resulting in incidental detection on radiographic examination [5]. CBCT has proven to be a valuable diagnostic tool, enhancing diagnostic accuracy and facilitating informed treatment planning for these asymptomatic lesions [6]. Moreover, incorporating CBCT evaluation into routine diagnostic protocols—even for non-endodontic purposes—can yield valuable incidental findings that provide significant insights into patient anatomy, pathology, and treatment outcomes, ultimately contributing to both clinical management and scientific knowledge. The prevalence of lingual root canals in mandibular incisors exhibits significant geographic, ethnic, age-related, and gender-based variability. Globally, the pooled prevalence of lingual canals in central incisors, lateral incisors, and canines is 21.9%, 26.0%, and 7.5%, respectively. Additionally, the prevalence of teeth with two confluent canals merging into a single foramen is 20.5% for central incisors, 24.1% for lateral incisors, and 4.9% for canines worldwide [7,8]. However, a study of a Caucasian population in Mexico revealed a notably lower prevalence, with reported rates of 5.7% for central incisors, 7.3% for lateral incisors, and 1.7% for canines, specifically for teeth with two confluent canals and one unique foramen [7,8]. Conversely, Vertucci’s Type III canal prevalence in permanent lower anterior teeth varies widely among populations. In European and Latin-American samples, central-incisor prevalence ranges from 8% to 55%, lateral-incisor from 9% to 52%, and canine from 4% to 8%, with bilaterality spanning 2.68–99.87% [9,10,11,12,13]. This inter-population variability underscores the importance of considering individual anatomical differences in clinical practice, as illustrated by our report. Figure 12 provides a comprehensive representation of Vertucci’s classification, illustrating the diverse root canal configurations and highlighting the Type III anatomy relevant to this report. The prevalence of lingual root canals in mandibular incisors exhibits significant geographic, ethnic, age-related, and gender-based variability. Globally, the pooled prevalence of lingual canals in central incisors, lateral incisors, and canines is 21.9%, 26.0%, and 7.5%, respectively. Additionally, the prevalence of teeth with two confluent canals merging into a single foramen is 20.5% for central incisors, 24.1% for lateral incisors, and 4.9% for canines worldwide [7,8]. However, a study of a Caucasian population in Mexico revealed a notably lower prevalence, with reported rates of 5.7% for central incisors, 7.3% for lateral incisors, and 1.7% for canines, specifically for teeth with two confluent canals and one unique foramen [7,8]. Conversely, Vertucci’s Type III canal prevalence in permanent lower anterior teeth varies widely among populations. In European and Latin-American samples, central-incisor prevalence ranges from 8% to 55%, lateral-incisor from 9% to 52%, and canine from 4% to 8%, with bilaterality spanning 2.68–99.87% [9,10,11,12,13]. This inter-population variability underscores the importance of considering individual anatomical differences in clinical practice, as illustrated by our report. Figure 12 provides a comprehensive representation of Vertucci’s classification, illustrating the diverse root canal configurations and highlighting the Type III anatomy relevant to this report.

**Figure 12 diagnostics-15-03157-f012:**
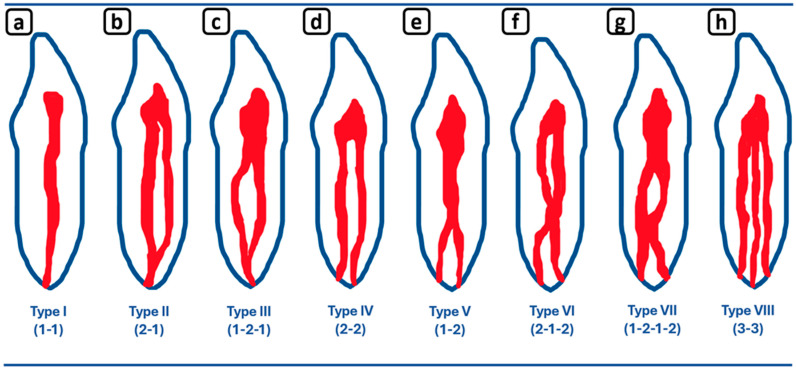
Vertucci’s classification of root canal configurations [4], adapted by the authors for the mandibular anterior sextant (incisors and canines), features eight types. (**a**) Type I. A single canal extends from the pulp chamber to the apex. (**b**) Type II. Two separate canals leave the pulp chamber and join short of the apex to form one canal. (**c**) Type III. One canal leaves the pulp chamber, divides into two within the root, and then merges to exit as one canal. (**d**) Type IV. Two separate and distinct canals extend from the pulp chamber to the apex. (**e**) Type V. One canal leaves the pulp chamber and divides short of the apex into two separate and distinct canals with separate apical foramina. (**f**) Type VI. Two separate canals leave the pulp chamber, merge in the body of the root, and redivide short of the apex to exit as two distinct canals. (**g**) Type VII. One canal leaves the pulp chamber, divides, and then rejoins within the body of the root, and finally redivides into two distinct canals short of the apex. (**h**) Type VIII. Three separate and distinct canals extend from the pulp chamber to the apex.

**Figure 13 diagnostics-15-03157-f013:**
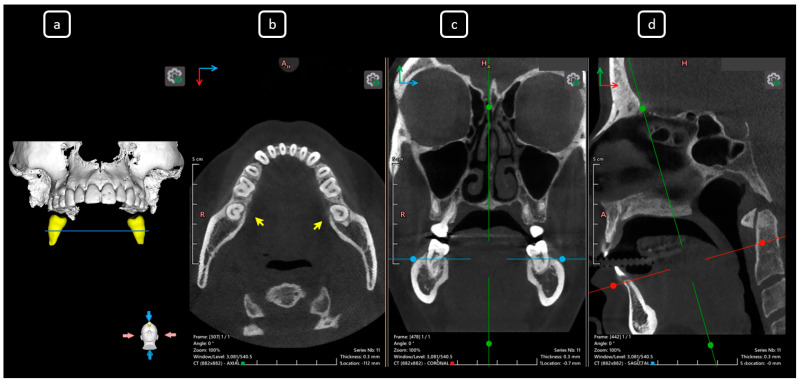
CBCT data with volumetric reconstruction and multiplanar views of the mandibular left and right second molars. (**a**) A 3D reconstruction of the second permanent mandibular molars (in yellow), the blue line indicates the axial slice level. (**b**) Axial section at the root cervical-third-level in teeth 3.7 and 4.7 [3] (yellow arrows) exhibiting C-shape configurations classified as C1 and C2 (Fan’s classification [14]) for teeth 3.7 and 4.7, respectively. (**c**) Coronal section. (**d**) Sagittal section. The [B] and [C] of mandibular root canals. Note that CBCT sections are not aligned with the individual longitudinal axes of teeth.

**Figure 14 diagnostics-15-03157-f014:**
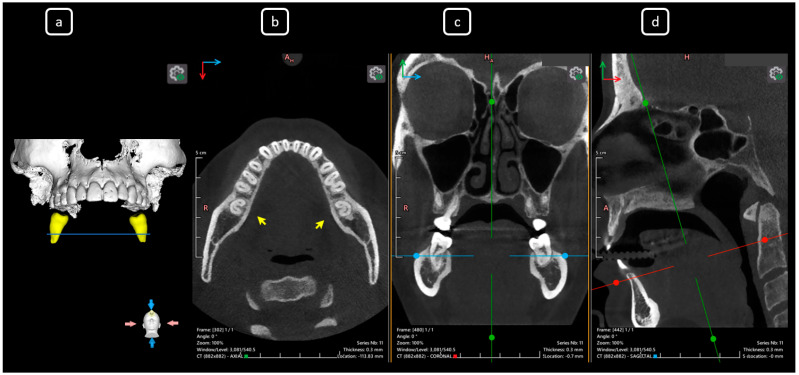
CBCT data with volumetric reconstruction and multiplanar views of the mandibular left and right second molars. (**a**) A 3D reconstruction of the second permanent mandibular molars (in yellow), the blue line indicates the axial slice level. (**b**) Axial section at the root middle-third level in teeth 3.7 and 4.7 [3] (yellow arrows), both exhibiting C-shape configurations classified as C2 according to Fan’s classification [14]. (**c**) Coronal section. (**d**) Sagittal section.

**Figure 15 diagnostics-15-03157-f015:**
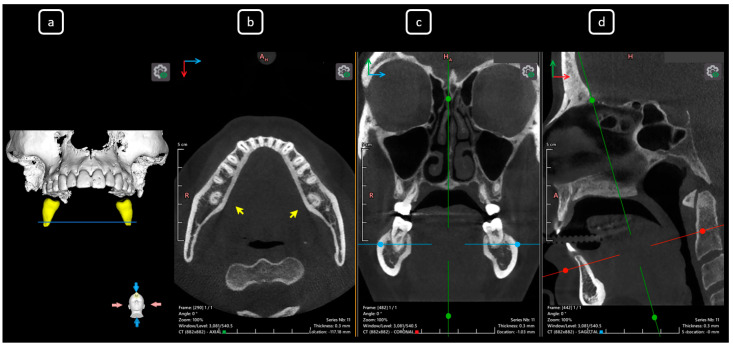
CBCT data with volumetric reconstruction and multiplanar views of the mandibular left and right second molars. (**a**) A 3D reconstruction of the second permanent mandibular molars (in yellow), the blue line indicates the axial slice level. (**b**) Axial section at the root apical-third level in teeth 3.7 and 4.7 [3] (yellow arrows), both exhibiting C-shape configurations classified as C3b according to Fan’s classification [14]. (**c**) Coronal section. (**d**) Sagittal section.

**Figure 16 diagnostics-15-03157-f016:**
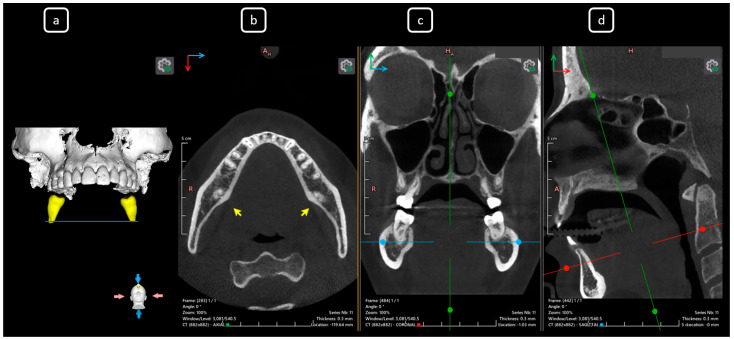
CBCT data with volumetric reconstruction and multiplanar views of the mandibular left and right second molars. (**a**) A 3D reconstruction of the second permanent mandibular molars (in yellow), the blue line indicates the axial slice level. (**b**) Axial section at the root apical level in teeth 3.7 and 4.7 [3] (yellow arrows), both exhibiting C-shape configurations classified as C3b according to Fan’s classification [14]. (**c**) Coronal section. (**d**) Sagittal section.

**Figure 17 diagnostics-15-03157-f017:**
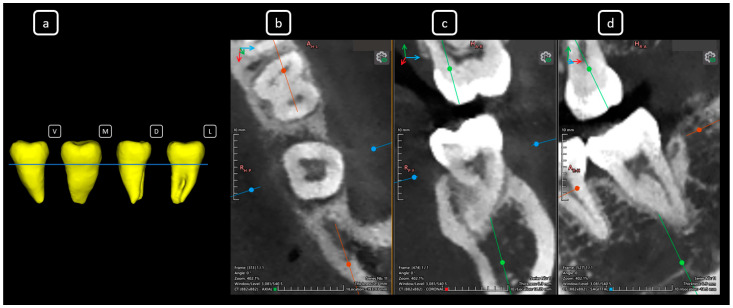
CBCT data with volumetric reconstruction and multiplanar views. (**a**) A 3D reconstruction of tooth 3.7 [3]. (V: vestibular, M: mesial, D: distal, L: lingual); the blue line indicates the axial slice level. (**b**) Axial section at the cervical level of tooth 3.7. (**c**) Coronal section. (**d**) Sagittal section. The [C] of mandibular root canals. Sections are oriented along the tooth’s longitudinal axis.

**Figure 18 diagnostics-15-03157-f018:**
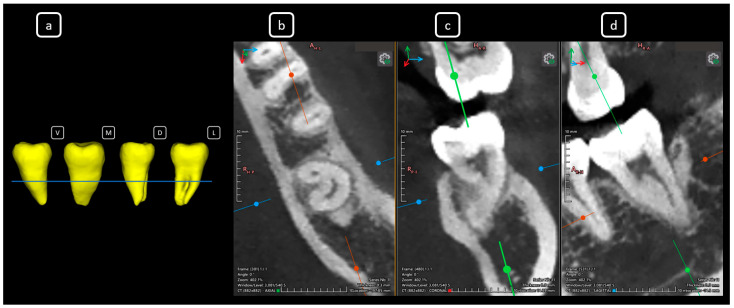
CBCT data with volumetric reconstruction and multiplanar views. (**a**) A 3D reconstruction of tooth 3.7 [3]. (V: vestibular, M: mesial, D: distal, L: lingual); the blue line indicates the axial slice level. (**b**) Axial section at the middle third of the root of tooth 3.7, revealing a C-shape configuration classified as C2 according to Fan’s classification [14]. (**c**) Coronal section. (**d**) Sagittal section.

**Figure 19 diagnostics-15-03157-f019:**
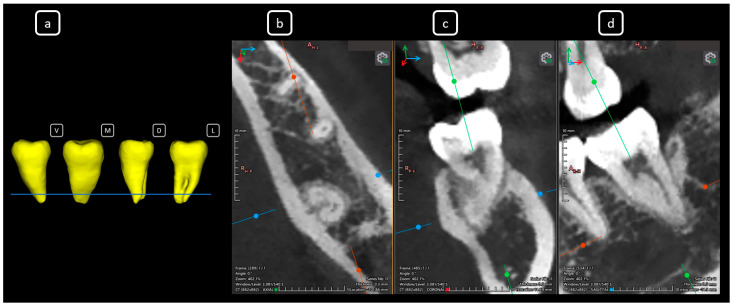
CBCT data with volumetric reconstruction and multiplanar views. (**a**) A 3D reconstruction of tooth 3.7 [3]. (V: vestibular, M: mesial, D: distal, L: lingual); the blue line indicates the axial slice level. (**b**) Axial section at the apical third of the root of tooth 3.7, revealing a C-shape configuration classified as C3b according to Fan’s classification [14]. (**c**) Coronal section. (**d**) Sagittal section.

**Figure 20 diagnostics-15-03157-f020:**
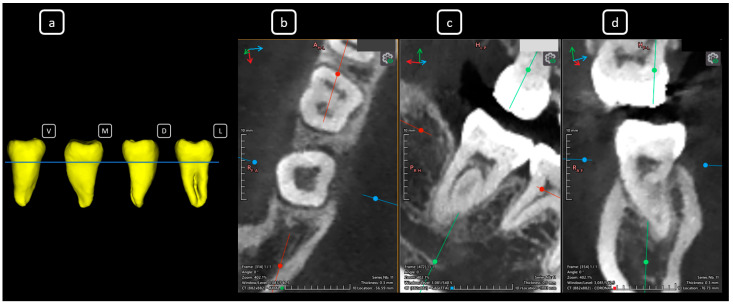
CBCT data with volumetric reconstruction and multiplanar views. (**a**) A 3D reconstruction of tooth 4.7 [3]. (V: vestibular, M: mesial, D: distal, L: lingual); the blue line indicates the axial slice level. (**b**) Axial section at the cervical level of tooth 4.7. (**c**) Coronal section. (**d**) Sagittal section. The [C] of mandibular root canals.

**Figure 21 diagnostics-15-03157-f021:**
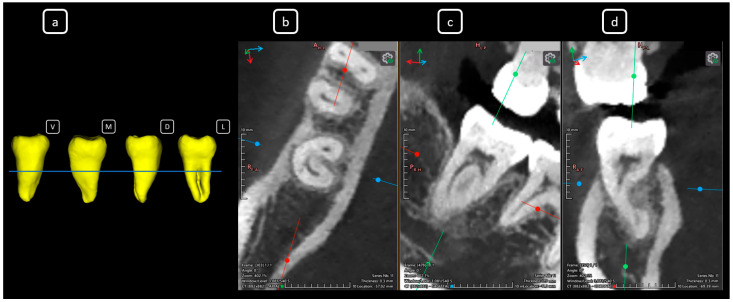
CBCT data with volumetric reconstruction and multiplanar views. (**a**) A 3D reconstruction of tooth 4.7 [3]. (V: vestibular, M: mesial, D: distal, L: lingual); the blue line indicates the axial slice level. (**b**) Axial section at the apical middle third of the root of tooth 4.7, revealing a C-shape configuration classified as C2 according to Fan’s classification [14]. (**c**) Coronal section. (**d**) Sagittal section.

**Figure 22 diagnostics-15-03157-f022:**
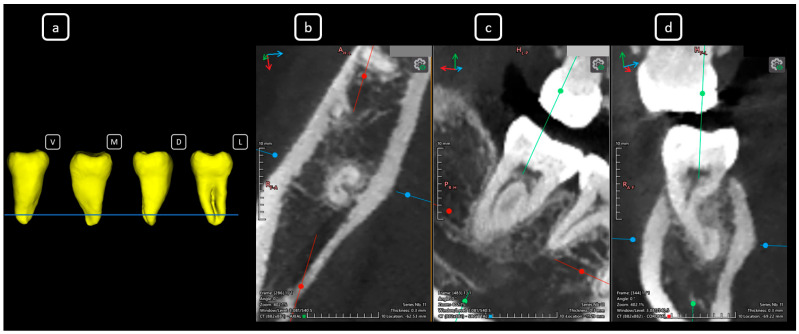
CBCT data with volumetric reconstruction and multiplanar views. (**a**) A 3D reconstruction of tooth 4.7 [3]. (V: vestibular, M: mesial, D: distal, L: lingual); the blue line indicates the axial slice level. (**b**) Axial section at the apical third of the root of tooth 3.7, revealing a C-shape configuration classified as C3b according to Fan’s classification [14]. (**c**) Coronal section. (**d**) Sagittal section. The C-shaped root canal configuration is a notable anatomical variation in mandibular second molars, with a reported global prevalence of 13.9% [15]. In Mexico, the prevalence is slightly higher at 14.2% [15]. Further characterization in specific studies has shown a prevalence of 10.2% (C1 type predominating in the coronal third (56%) and C2 type occurring in 44.1% of cases in the middle third) [16]. Moreover, the prevalence of C-shaped root canal morphology, in permanent mandibular second molars, varies markedly among populations. In Asia, a Chinese investigation reported a prevalence of 38.6% with 81.3% bilaterality [17]; Korean studies found 39.8% (71.6% bilateral) [18] and 36.8% (75.3% bilateral) [19], respectively, while a Xinjiang study observed 29.2% (43.3% bilateral) [20]. In the Middle East, a Turkish report gave 8.9% (6.3% bilateral) [21]; a Saudi study 9.1% (46.2% bilateral) [22]; a United Arab Emirates study 17.9% (71.7% bilateral) [23]; a Jordanian study 12.0% (28.9% bilateral) [24]; and an Iraqi study 17.4% (64.1% bilateral) [25]. In Latin America, a Brazilian study found 15.3% (31.7% bilateral) [26], and an Ecuadorian study reported 28% (63% bilateral) [27]. In Europe and Oceania, an Israeli study reported 4.6% (45% bilateral) [28]; an Australian study, 13% (95% bilateral) [29]; a Lebanese study, 9.1% (60% bilateral) [30]; and a Spanish study, 10.2% (54.8% bilateral) [16].

This wide range of prevalences and bilaterality proportions underscores the complexity of C-shape root canal morphology and the need for thorough diagnostic evaluation in clinical practice. Figure 23 illustrates the C-shape classification system, offering a clear visual of its configurations [14].

## Figures and Tables

**Figure 23 diagnostics-15-03157-f023:**
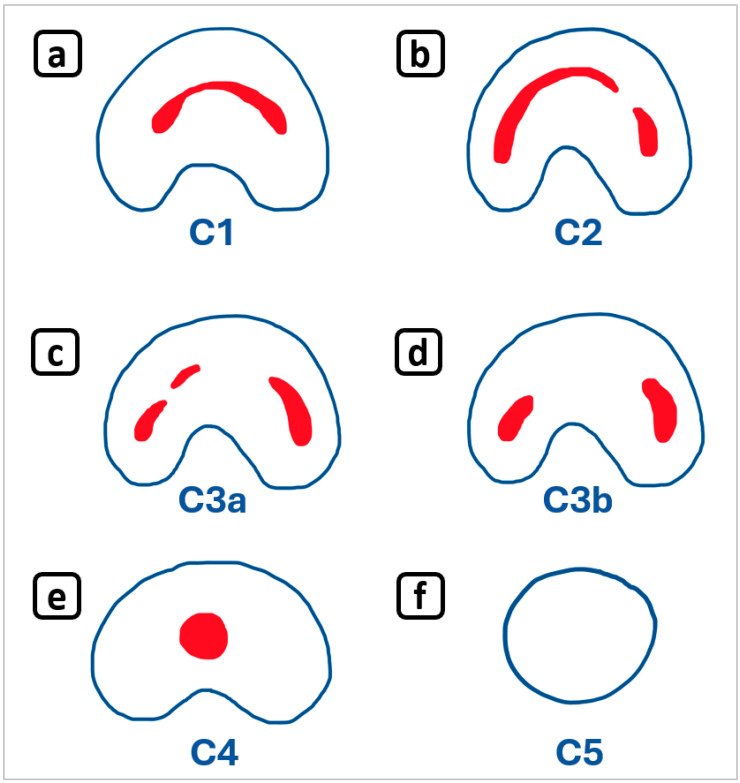
Classification system of C-shaped root canal anatomy configurations by Fan et al. [14] for the mandibular second molar, illustrating the five types, for comparison with the anatomical findings in this report. (**a**) Category C1: The shape was an uninterrupted “C” with no separation or division. (**b**) Category C2: The canal shape resembled a semicolon resulting from a discontinuation of the “C” outline. (**c**) Category C3: Two or three separate canals (**c**,**d**). (**e**) Category C4: Only one round or oval canal in that cross-section. (**f**) Category C5: No canal lumen could be observed (confined to the apical area). Figure adapted by the authors based on data from [14]. The implementation of cone-beam computed tomography (CBCT) in dentistry in the late 1990s marked a radical change and a significant advance in dental imaging [31,32]. CBCT equipment includes software that enables clinicians to visualize, reconstruct, and accurately measure structures such as bone, teeth, and the upper airway [33]. Images are stored and managed in DICOM format, the international standard for the exchange and management of digital medical images [34]; this facilitates visualization of large data sets with specialized DICOM viewers [35]. Moreover, CBCT has proven useful for three-dimensional evaluation of the airways, revealing the intricate relationship between dental anatomy and upper-airway morphology, and it can lead to the incidental discovery of unusual root -canal anatomies [36,37,38,39,40]. Large-field-of-view CBCT scans enable exploration and identification of structures in a single exposure, correlating different fields of medicine using the same imaging study for diagnostic, therapeutic, or educational purposes [41]. In contrast, small-field-of-view CBCT is recommended for complex endodontic cases, reducing scatter and improving image quality in confined spaces such as root canals [42,43]. It is worth noting that encountering unusual root canal images on CBCT scans requires specific academic and clinical training in identifying internal and external root anatomy [44,45]. The relative rarity of such findings can be attributed to the fact that CBCT scans are not exclusive to dentistry and are not routine examinations in either medicine or dentistry [44]. Professional organizations, such as the American Association of Endodontists (AAE) and the European Society of Endodontology (ESE), provide guidelines for the use of CBCT in endodontics, recommending its application in complex cases where 2D imaging is insufficient for diagnosis, particularly in cases of non-healing root canals, trauma, or suspected extra-anatomical root canals [46,47]. According to these guidelines, CBCT should not be used for routine screening but rather as a valuable tool to aid in diagnosis and treatment planning in select cases [48]. By adhering to these guidelines and leveraging the capabilities of CBCT technology, clinicians can optimize patient care and improve treatment outcomes. Incidental findings on CBCT scans performed for non-endodontic purposes—such as upper-airway evaluation—can reveal complex root -canal anatomies with important implications for treatment planning and execution. Detecting these variations guides modifications to access-cavity design, instrumentation, and irrigation protocols, ultimately enhancing efficacy and patient outcomes. Thorough evaluation of CBCT images is therefore essential. Vertucci Type III configurations require wider access and careful negotiation of buccal and lingual canals, together with advanced instrumentation, irrigation, and obturation. Because of the increased risk of failure during the endodontic phases, the prognosis is guarded [49,50]. C-shaped canals demand a tailored access design, accurate anatomical identification, and meticulous instrumentation, coupled with fluent irrigation to eliminate organic and bacterial debris and to facilitate adequate obturation. Although challenging, a favorable prognosis can be achieved with proper management and adequate final restorations [51,52]. Careful planning, a thorough understanding of anatomy, and the judicious use of advanced technologies are crucial for optimizing treatment outcomes, underscoring the importance of a meticulous approach in managing complex root -canal anatomies. **Implications for Practice:** The identification of complex root canal anatomy has significant implications for endodontic planning. Suspicion of complex anatomy may be informed by knowledge of anatomical variations prevalent in specific populations, familiarity with tooth morphologies, and careful evaluation of 2D radiographic features, indicating the need for targeted CBCT imaging. CBCT imaging facilitates a comprehensive understanding of each tooth’s unique anatomy, enabling clinicians to develop targeted treatment plans. A modified access-cavity design is essential to minimize excessive tooth removal and optimize visibility. A tailored scouting sequence guides exploration, navigates intricate canal anatomy, reduces procedural errors, and allows ac-curate working-length determination. Isthmus management is critical for thorough cleaning and shaping. Effective placement and removal of intracanal medicament is vital, especially in teeth with C-shaped canals, Vertucci Type III configurations, or multiple canal systems. A customized obturation technique ensures complete 3-D filling. Consequently, omitting CBCT imaging may increase the risk of overlooking anatomical variations. Finally, to our knowledge, this is the first report describing the co-occurrence of double root canals in the mandibular anterior sextant and bilateral C-shaped permanent second mandibular molars. This finding adds to the limited literature on rare anatomical variations and underscores the importance of CBCT imaging for accurate diagnosis and treatment planning. The convergence of (A) anterior atypical root-canal variations, (B) bilateral symmetry, (C) C-shaped morphology in mandibular second molars, and (D) a “1-2-1” canal configuration is rare, but its presence merits consideration. This poses a significant diagnostic and clinical challenge, underscoring the need for clinicians to be cognizant of such complexities. Multidisciplinary interpretation of CBCT data, involving collaboration between radiologists, clinicians, and endodontic specialists, can enhance diagnostic accuracy and optimize patient management with appropriate patient records. Documenting and sharing unusual cases [53,54,55] can facilitate the detection of similar cases, sensitizing readers to rare or previously unreported findings and contributing to the advancement of medical knowledge.

## Data Availability

Data are available from the corresponding author upon reasonable request and with the permission of the Faculty of Dentistry, Universidad Juárez del Estado de Durango, Mx. The CBCT images in DICOM (Digital Imaging and Communication in Medicine) format were obtained with the Newtom VGI Evo device (Newtom, Imola, Italy), with a window of 24 × 19 cm and exposure time of 110 kV, 2 mA, and 15 s. The CBCT DICOM data were analyzed using a combination of two free-access software tools. Weasis MPR Viewer (version 4.6.0) was utilized for multi-planar reconstruction, allowing for detailed examination of the data set in multiple planes. Additionally, Blue Sky Plan^®^ (version 4.13) dental imaging software was employed to render high-quality 3D volumetric images, providing a comprehensive visualization of the anatomical structures. Both software tools seamlessly integrated the DICOM data, ensuring accurate rendering and analysis. Image analysis was performed by an endodontic specialist (E.G.-T.).

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
