# Peer review of "Imaging Findings of Clinical Significance in Endodontics During Cone Beam Computed Tomography Scanning of the Upper Airway—The Anterior, Bilateral, C-Shaped, Dual of Mandibular Root Canals: A Brief Case Report"

_diagnostics, 2025, doi:10.3390/diagnostics15243157_

Round 1

Reviewer 1 Report (New Reviewer)

Comments and Suggestions for Authors

Dear authors,

I read with interest your manuscript titled "Imaging findings of clinical significance in endodontics during cone beam computed tomography scanning of the upper airway. The A[nterior], B[ilateral], C[shape], D[ual] of mandibular root canals: a brief case report."

It is a well presented imaging series of various anatomic variations of root canals of mandibular teeth. Despite that, I do not find a specific interest in this, as anatomical variations are well described in the literature, and the significance of CBCT is well established. In addition, It would be beneficial to establish a causation of performing a non-endodontic CBCT and studying effectively incidental anatomical variations, which do not have an clinical meaning if no endodontic treatment is to be performed. All in all, apart from the interest in the imaging series, I do not find a useful meaning on this Interesting Images manuscript.

Author Response

We are pleased to resubmit our manuscript, “Imaging findings of clinical significance in endodontics during cone beam computed tomography scanning of the upper airway. The A[nterior], B[ilateral], C[shape], D[ual] of mandibular root canals”, for consideration for publication in Diagnostics Journal-MDPI, following the second round of review.

This manuscript was originally submitted with ID diagnostics-3808872 and is now being resubmitted with ID diagnostics-3994260. We appreciate the reviewers' time and expertise in evaluating our work, and we are grateful for the opportunity to revise and resubmit.

Attached is a detailed, point-by-point response to the peer reviewers' comments, outlining the specific changes made to address their feedback. We have thoroughly addressed all comments and suggestions from the three reviewers, and believe the manuscript has been substantially strengthened as a result.

Comment 1:

  • Dear authors, I read with interest your manuscript titled "Imaging findings of clinical significance in endodontics during cone beam computed tomography scanning of the upper airway. The A[nterior], B[ilateral], C[shape], D[ual] of mandibular root canals: a brief case report."

It is a well presented imaging series of various anatomic variations of root canals of mandibular teeth. Despite that, I do not find a specific interest in this, as anatomical variations are well described in the literature, and the significance of CBCT is well established. In addition, It would be beneficial to establish a causation of performing a non-endodontic CBCT and studying effectively incidental anatomical variations, which do not have an clinical meaning if no endodontic treatment is to be performed. All in all, apart from the interest in the imaging series, I do not find a useful meaning on this Interesting Images manuscript.

Response:

  • We appreciate the reviewers' feedback and acknowledge that anatomical variations are well-documented. However, this case's unique combination of multiple rare variations (Vertucci's type III and bilateral C-shaped mandibular second molars) makes it a valuable addition to the literature, serving as an educational example for clinicians and contributing to the ongoing understanding of root canal anatomy.

We agree that the clinical implications of incidentally detected anatomical variations are crucial. In this report, documenting these variations is valuable for the patient's record, alerting clinicians to potential complexities if future endodontic treatment is needed. This highlights CBCT's broader utility in informing patient care beyond the initial scan indication, particularly in guiding diagnostic approaches and treatment planning in endodontics.

Following the first round of reviews, we have revised the manuscript to improve clarity and added referential background to strengthen its context. In response to the second round of reviews, we have incorporated further revisions to address the peer reviewers' suggestions, enhancing the manuscript's overall quality and validity. We respectfully submit that this report meets the criteria for an Interesting Images manuscript, as it presents a novel finding with significant educational value.

Our approach underscores comprehensive patient management, recognizing that we treat patients, not just teeth. The CBCT study, although performed for upper airway evaluation, provided valuable information about the patient's dental anatomy, highlighting the benefits of integrating imaging findings into overall patient care. This holistic approach can inform future treatment decisions and improve patient outcomes.

This case exemplifies CBCT's integral role in modern endodontic diagnostics, enabling clinicians to accurately identify and understand complex anatomical variations, leading to more effective and personalized treatment strategies.

In summary, our manuscript embodies Novelty, Scientific Soundness, a Well-Structured presentation, and Alignment with Diagnostics journal's guidelines, making it a valuable contribution to diagnostic imaging and its applications in patient care.

We would like to extend our sincere appreciation for the rigorous and impartial revision of our manuscript. Your insightful comments and constructive feedback have been instrumental in strengthening the quality and evidentiary foundation of this work, and we are grateful for the time and expertise you dedicated to this process.

Reviewer 2 Report (New Reviewer)

Comments and Suggestions for Authors

The manuscript presents a unique and rare set of anatomical variations of mandibular roots (A, B, C, D), which were discovered incidentally during a CBCT scan of the upper respiratory tract. The topic is educationally valuable and aligns well with the Interesting Images section of the Diagnostics journal. The imaging quality is rich and highly detailed, and the literature referenced is up to date. The text requires revisions.

  • The abstract is too lengthy and includes controversial elements.
  • The descriptive and epidemiological sections are excessively detailed. The manuscript has extensive information on the frequency of individual anatomical variants. I recommend shortening the epidemiological section by 60–70%, retaining only the essential data necessary for understanding the rarity of this case.
  • Additionally, there is an excessive number of figures and some repetitions. The article contains over 25 figures, many of which are quite similar. I suggest limiting the number of illustrations to a maximum of 10-12 and combining duplicate.
  • Technical descriptions are duplicated. The phrase, "Note that CBCT sections are not aligned with the individual longitudinal axes of teeth," appears multiple times. This information should be included only once, such as in the legend of the first figure.
  • The explanation of the "A–B–C–D" concept is insufficient. The acronym is only introduced at the end of the manuscript, making it unclear for the reader. It is recommended to define the A–B–C–D concept in the introduction, preferably within the first paragraphs.
  • Figures and legends need to be merged and shortened. Some legends are overly lengthy and repeat information already found in the text. It's advisable to condense them to essential elements and combine duplicate sets of images into more readable panels.
  • There are minor linguistic and stylistic issues. The text contains repetitions in certain areas (e.g., "notably," "furthermore"), lengthy sentences, and a writing style more characteristic of a review than a case report. Editing for clarity and conciseness is recommended.

Author Response

We are pleased to resubmit our manuscript, “Imaging findings of clinical significance in endodontics during cone beam computed tomography scanning of the upper airway. The A[nterior], B[ilateral], C[shape], D[ual] of mandibular root canals”, for consideration for publication in Diagnostics Journal-MDPI, following the second round of review.

This manuscript was originally submitted with ID diagnostics-3808872 and is now being resubmitted with ID diagnostics-3994260. We appreciate the reviewers' time and expertise in evaluating our work, and we are grateful for the opportunity to revise and resubmit.

Attached is a detailed, point-by-point response to the peer reviewers' comments, outlining the specific changes made to address their feedback. We have thoroughly addressed all comments and suggestions from the three reviewers, and believe the manuscript has been substantially strengthened as a result.

Thank you for your time and consideration. 

Comments and Suggestions for Authors, Reviewer 2.

Comment 1:

  • The abstract is too lengthy and includes controversial elements.

Response:

  • We appreciate your feedback on our manuscript. Following the initial review, we were guided to adhere to the CARE Guidelines, which led to the original abstract's length. We ensured the submitted version remained within the 200-word limit (171). In response to your suggestion, we have revised the abstract to make it more concise, focusing on key points and removing potentially controversial elements.

The revised abstract now reads:

“Cone beam computed tomography (CBCT) is a valuable diagnostic tool for evaluating the upper airway and maxillofacial region. This report demonstrates the clinical value of CBCT in identifying significant anatomical variations in endodontics, incidentally detected on a non-endodontic CBCT scan. A 23-year-old female patient underwent CBCT imaging at the Faculty of Dentistry-UJED to evaluate her upper airway. CBCT imaging revealed a unique, complex, and unusual anatomy of mandibular root canals, characterized by Vertucci's type III root canals in the anterior sextant and co-occurrence of bilateral C-shaped mandibular second molars (type C2 according to Fan's classification). No therapeutic interventions were initiated due to the patient's asymptomatic status. CBCT imaging is a valuable tool for integrated diagnostic approaches, underscoring its role in thorough patient management. The integration of multidisciplinary interpretation of CBCT data can enhance diagnostic accuracy and optimize patient records and management, emphasizing the importance of collaborative efforts between radiologists, clinicians, and endodontists. Documenting and sharing such findings can increase awareness of rare anatomical variations, facilitating detection and contributing to medical knowledge.”

Comment 2:

  • The descriptive and epidemiological sections are excessively detailed. The manuscript has extensive information on the frequency of individual anatomical variants. I recommend shortening the epidemiological section by 60–70%, retaining only the essential data necessary for understanding the rarity of this case.

Response:

  • We appreciate your suggestion to shorten the epidemiological section. In the initial review, we were asked to include detailed tables with epidemiological information on individual and bilateral presentations of these anatomical variants in various populations worldwide. Adding this information was a key revision request, and its absence was a major reason for the manuscript's initial rejection.

Given the contrasting feedback, we have carefully revised the epidemiological section to strike a balance between providing essential context and avoiding excessive detail. We have retained key data highlighting the rarity of this case and summarized the remaining information in a concise manner.

The revised section now focuses on the most relevant epidemiological aspects, ensuring the manuscript meets the journal's requirements. Changes are highlighted in yellow

Comment 3:

  • Additionally, there is an excessive number of figures and some repetitions. The article contains over 25 figures, many of which are quite similar. I suggest limiting the number of illustrations to a maximum of 10-12 and combining duplicate.

Response:

  • We appreciate your suggestion to reduce the number of figures. In the initial review, we were asked to enhance imaging sequences with detailed sections over the root length to better describe the findings, which was a key revision request. This led to the inclusion of comprehensive images to accurately represent the case, addressing one of the main reasons for the manuscript's initial rejection.

In this revision, we have carefully selected representative images of interest, balancing clarity with the goal of avoiding redundancy. Even when there are no repetitive images, everyone has its own reason to be included, providing unique insights or illustrating specific aspects of the case.

We believe the revised set of figures effectively supports the manuscript's objectives and adequately describes the findings in a scientific manner. Notably, the Diagnostics MDPI Journal guidelines encourage the submission of Interesting Images, and the number of images is at the discretion of the author. We have revised the manuscript to meet the journal's requirements while addressing your concerns.

Comment 4:

  • Technical descriptions are duplicated. The phrase, "Note that CBCT sections are not aligned with the individual longitudinal axes of teeth," appears multiple times. This information should be included only once, such as in the legend of the first figure.

 Response:

  • We appreciate your observation regarding the duplicated technical descriptions. The phrase "Note that CBCT sections are not aligned with the individual longitudinal axes of teeth" was included in multiple figure legends to ensure clarity, as each image is described in detail. Given the manuscript's focus on describing imaging findings and their uniqueness, CBCT frames are presented in various orientations: some aligned with individual longitudinal axes, others showing the anterior sextant or bilateral posterior molar sections.

In response to your suggestion, we've revised the manuscript to include this information only once, typically in the legend of the first figure, and improved the image legends for clarity and concision. We believe this revision maintains scientific accuracy and descriptive quality while avoiding unnecessary repetition.

Comment 5:

  • The explanation of the "A–B–C–D" concept is insufficient. The acronym is only introduced at the end of the manuscript, making it unclear for the reader. It is recommended to define the A–B–C–D concept in the introduction, preferably within the first paragraphs.

Response:

  • We appreciate your suggestion to clarify the "A-B-C-D" concept. In response, we've expanded the introduction to define this concept, highlighting the convergence of complex anatomical variations.

Now reads as follows:

“Such variability is exemplified by the A-B-C-D concept, which embodies a complex paradigm of anatomical variability, where multiple, seemingly disparate anomalies converge to create a singular clinical entity. This intricate constellation of variations is characterized by the following key features:

Anterior atypical anatomy (A): Mandibular incisors and canines may display unusual root‑canal configurations, including supplementary canals, aberrant trajectories, or unexpected apical terminations.

Bilateral symmetry (B): Identical anomalies are mirrored on contralateral sides of the dental arch, which can be related to genetic or environmental influences on morphogenesis.

C-shaped morphology (C): Mandibular second molars display a complex, C-shaped canal system, wrapping around the distal root and creating a formidable challenge for chemomechanical debridement and obturation.

Dual nature of root canal systems (D): The "1-2-1" configuration exemplifies the intricate variability of root canal anatomy, featuring a single canal that bifurcates and reunites, creating a complex network of channels, anastomoses, and apical deltas.

This A-B-C-D constellation of anatomical variations demands a high degree of diagnostic acumen, technical expertise, and adaptability from the clinician, as well as a profound appreciation for the intricate complexity and variability of the human dentition. The co-occurrence of these variations is exceedingly rare, emphasizing the need for thorough CBCT evaluation and clinician awareness in identifying such complex anatomical presentations. These variations defy conventional norms and challenge traditional endodontic dogma.”

Comment 6:

  • Figures and legends need to be merged and shortened. Some legends are overly lengthy and repeat information already found in the text. It's advisable to condense them to essential elements and combine duplicate sets of images into more readable panels.

Response:

As mentioned earlier, the figures and legends were revised to include only essential elements, combining duplicate (if any) sets of images into more readable panels, and avoiding repetition of information already found in the text, while maintaining a scientific and informative tone.

The number and description of figures were modified to reflect this, ensuring that the legends are concise and focused on key findings, without compromising the scientific accuracy and clarity of the information presented.

Comment 7:

  • There are minor linguistic and stylistic issues. The text contains repetitions in certain areas (e.g., "notably," "furthermore"), lengthy sentences, and a writing style more characteristic of a review than a case report. Editing for clarity and conciseness is recommended.

Response:

  • Thank you for the feedback. We´ve revised the text to improve clarity and conciseness, avoiding repetitive language and lengthy sentences.

Regarding the writing style, the Diagnostics MDPI Journal guidelines for Interesting Images specify that no regular manuscript text should be included (introduction/methods/results/discussion). Instead, images are accompanied by detailed legends with no length restriction, and reference citations appear in the legends. This contributed to the text's length and style. We were previously indicated to apply CARE Guidelines to ensure scientific soundness, following the journal's specific template for Interesting Images.

We'll ensure the manuscript adheres to the guidelines, improves clarity and concision, and incorporates your valuable suggestions to enhance quality.

We would like to extend our sincere appreciation for the rigorous and impartial revision of our manuscript. Your insightful comments and constructive feedback have been instrumental in strengthening the quality and evidentiary foundation of this work, and we are grateful for the time and expertise you dedicated to this process.

Reviewer 3 Report (New Reviewer)

Comments and Suggestions for Authors  

The manuscript presents a concise and visually strong case report illustrating the incidental discovery of complex mandibular root canal anatomy during a CBCT scan primarily obtained for upper airway assessment. The topic is relevant and educational for both radiologists and endodontists, emphasizing the diagnostic potential of CBCT beyond its initial indication.

Overall, the manuscript is scientifically sound, well-illustrated, and supported by an updated and comprehensive literature review. However, several minor revisions would enhance clarity, consistency, and scientific precision:

  1. Abstract and Introduction:

    • Clarify the clinical context of the CBCT scan and highlight why this incidental finding is noteworthy in comparison with existing prevalence data.

    • Shorten the introductory background slightly; the focus should remain on the diagnostic and clinical relevance rather than extensive anatomical theory.

  2. Figures and Captions:

    • The figures are excellent in quality. However, ensure that each image has a concise and self-contained caption describing the key finding (e.g., specify which Vertucci or Fan classification type is demonstrated).

    • Verify the figure numbering sequence—some captions may overlap in numbering or description.

  3. Discussion:

    • Strengthen the clinical discussion by expanding on how such anatomic variations could alter endodontic access design, instrumentation strategy, or prognosis.

    • The section titled Clinical relevance is useful but can be rewritten more cohesively as a short “Implications for Practice” paragraph integrating the bullet points into a narrative form.

  4. Language and Style:

    • The English language is fluent, but a few overly long sentences in the introduction and discussion could be split for better readability.

    • Replace phrases like “this brief case report highlights” with more formal scientific transitions such as “this case demonstrates” or “the findings suggest”.

  5. Conclusion:

    • Emphasize the message that CBCT—even when performed for non-endodontic reasons—can yield clinically meaningful incidental findings.

    • Suggest that multidisciplinary interpretation of CBCT data (radiology and endodontics) improves diagnostic precision and patient management.

After these refinements, the manuscript will be well suited for publication in the Interesting Images section of Diagnostics, offering both visual and didactic value to the readership.

Author Response

We are pleased to resubmit our manuscript, “Imaging findings of clinical significance in endodontics during cone beam computed tomography scanning of the upper airway. The A[nterior], B[ilateral], C[shape], D[ual] of mandibular root canals”, for consideration for publication in Diagnostics Journal-MDPI, following the second round of review.

This manuscript was originally submitted with ID diagnostics-3808872 and is now being resubmitted with ID diagnostics-3994260. We appreciate the reviewers' time and expertise in evaluating our work, and we are grateful for the opportunity to revise and resubmit.

Attached is a detailed, point-by-point response to the peer reviewers' comments, outlining the specific changes made to address their feedback. We have thoroughly addressed all comments and suggestions from the three reviewers, and believe the manuscript has been substantially strengthened as a result.

Thank you for your time and consideration. 

Comments and Suggestions for Authors, Reviewer 3.

Comment 1:

  • Abstract and Introduction:

Clarify the clinical context of the CBCT scan and highlight why this incidental finding is noteworthy in comparison with existing prevalence data.

Response:

  • Thank you for your suggestion to clarify the clinical context of the CBCT scan. We've ensured that the abstract now includes this information, stating: "A 23-year-old female patient underwent CBCT imaging at the Faculty of Dentistry-UJED to evaluate her upper airway. CBCT imaging revealed a unique, complex, and unusual anatomy of mandibular root canals...", as we've noted in the manuscript, "The scan was performed for upper airway evaluation as requested by her physician; the patient was asymptomatic for pain or symptoms related to the area." This can provide clear context for the CBCT scan, highlighting the incidental nature of the finding, and addresses your comment.

We believe this addresses your comment and enhances the manuscript's clarity.

Comment 2:

  • Shorten the introductory background slightly; the focus should remain on the diagnostic and clinical relevance rather than extensive anatomical theory.

Response:

  • We appreciate your suggestions and those of the other peer reviewers. We've revised the manuscript to improve clarity and focus, shortening the introductory background to emphasize the diagnostic and clinical relevance of the finding, as suggested. The upper airway evaluation was the main indication for the CBCT, and we took advantage of the images to examine the maxillofacial and dental areas, revealing the complex anatomy of the mandibular root canals.

In response to the reviewers' comments, we've made targeted revisions to enhance the manuscript's quality and adherence to the journal's guidelines, balancing brevity with essential information. We've ensured that all suggestions have been addressed, resulting in a more concise and focused manuscript. We note that some suggestions from different reviewers may have been contradictory; however, we've made an effort to fulfill all suggestions and balance the feedback to improve the manuscript.

Comment 3:

  • Figures and Captions:

The figures are excellent in quality. However, ensure that each image has a concise and self-contained caption describing the key finding (e.g., specify which Vertucci or Fan classification type is demonstrated).

Response:

  • Thank you for your positive feedback on the quality of the figures. We have revised the captions to include concise and self-contained descriptions of the key findings, including the specific Vertucci and Fan classification types demonstrated in each image where these were missing.

Comment 4:

  • Verify the figure numbering sequence—some captions may overlap in numbering or description.

Response:

  • Thank you for pointing this out. We have reviewed the figure numbering sequence and verified that all figures are correctly numbered and captioned. We have ensured that there are no overlaps in numbering or description.

Comment 5:

  • Discussion:

Strengthen the clinical discussion by expanding on how such anatomic variations could alter endodontic access design, instrumentation strategy, or prognosis.

Response:

  • Thank you for pointing this out. We have incorporated the suggested edits, and the revised paragraph now reads:

“Incidental findings on CBCT scans performed for non‑endodontic purposes—such as upper‑airway evaluation—can reveal complex root‑canal anatomies with important implications for treatment planning and execution. Detecting these variations guides modifications to access‑cavity design, instrumentation, and irrigation protocols, ultimately enhancing efficacy and patient outcomes. Thorough evaluation of CBCT images is therefore essential.

Vertucci Type III configurations require wider access and careful negotiation of buccal and lingual canals, together with advanced instrumentation, irrigation, and obturation. Because of the increased risk of failure during the endodontic phases, the prognosis is guarded [49,50].

C‑shaped canals demand a tailored access design, accurate anatomical identification, and meticulous instrumentation, coupled with fluent irrigation to eliminate organic and bacterial debris and to facilitate adequate obturation. Although challenging, a favorable prognosis can be achieved with proper management and adequate final restorations [51,52].

Careful planning, a thorough understanding of anatomy, and the judicious use of advanced technologies are crucial for optimizing treatment outcomes, underscoring the importance of a meticulous approach in managing complex root‑canal anatomies.

Comment 6:

  • The section titled Clinical relevance is useful but can be rewritten more cohesively as a short “Implications for Practice” paragraph integrating the bullet points into a narrative form.

Response:

  • Thank you for your feedback. We have incorporated the suggested edits, and the revised paragraph now reads:

Implications for Practice

The identification of complex root canal anatomy has significant implications for endodontic planning. Suspicion of complex anatomy may be informed by knowledge of anatomical variations prevalent in specific populations, familiarity with tooth morphologies, and careful evaluation of 2D radiographic features, indicating the need for targeted CBCT imaging. CBCT imaging facilitates a comprehensive understanding of each tooth's unique anatomy, enabling clinicians to develop targeted treatment plans.

A modified access‑cavity design is essential to minimize excessive tooth removal and optimize visibility. A tailored scouting sequence guides exploration, navigates intricate canal anatomy, reduces procedural errors, and allows accurate working‑length determination. Isthmus management is critical for thorough cleaning and shaping.  Effective placement and removal of intracanal medicament is vital, especially in teeth with C‑shaped canals, Vertucci Type III configurations, or multiple canal systems. A customized obturation technique ensures complete 3‑D filling. Consequently, omitting CBCT imaging may increase the risk of overlooking anatomical variations.”

Comment 7:

  • Language and Style:

The English language is fluent, but a few overly long sentences in the introduction and discussion could be split for better readability.

Response:

  • Thank you for your valuable feedback. We appreciate your suggestions and have made the necessary revisions to improve the manuscript.

We have revised the introduction and discussion sections to improve sentence structure and readability. Specifically, we have split long sentences and rephrased them for better clarity.

We would like to note that the Diagnostics MDPI journal's format guidelines for "Interesting Images" style do not allow for standard manuscript sections, which may contribute to the text appearing lengthy. We have worked to optimize the text within the constraints of the format, but appreciate your understanding in this regard.

Comment 8:

  • Replace phrases like “this brief case report highlights” with more formal scientific transitions such as “this case demonstrates” or “the findings suggest”.

Response:

  • Regarding the language and style, we have replaced informal phrases with more formal scientific transitions. For example, 'this brief case report highlights' has been revised to 'this case demonstrates' as suggested.

Comment 9:

  • Conclusion:

Emphasize the message that CBCT—even when performed for non-endodontic reasons—can yield clinically meaningful incidental findings.

Suggest that multidisciplinary interpretation of CBCT data (radiology and endodontics) improves diagnostic precision and patient management.

Response:

  • Thank you for your valuable feedback. We have revised the conclusion to highlight this key point and the revised paragraph now reads:

“Finally, to our knowledge, this is the first report describing the co‑occurrence of double root canals in the mandibular anterior sextant and bilateral C‑shaped permanent second mandibular molars. This finding adds to the limited literature on rare anatomical variations and underscores the importance of CBCT imaging for accurate diagnosis and treatment planning.

The convergence of (A) anterior atypical root‑canal variations, (B) bilateral symmetry, (C) C‑shaped morphology in mandibular second molars, and (D) a “1‑2‑1” canal configuration is rare, but its presence merits consideration. This poses a significant diagnostic and clinical challenge, underscoring the need for clinicians to be cognizant of such complexities.

Multidisciplinary interpretation of CBCT data, involving collaboration between radiologists, clinicians, and endodontic specialists, can enhance diagnostic accuracy and optimize patient management with appropriate patient records. Documenting and sharing unusual cases [53-55] can facilitate the detection of similar cases, sensitizing readers to rare or previously unreported findings and contributing to the advancement of medical knowledge.”

we would like to extend our sincere appreciation for the rigorous and impartial revision of our manuscript. Your insightful comments and constructive feedback have been instrumental in strengthening the quality and evidentiary foundation of this work, and we are grateful for the time and expertise you dedicated to this process.

Round 2

Reviewer 1 Report (New Reviewer)

Comments and Suggestions for Authors

Thank you for revising the manuscript. I do not have any further comments.

This manuscript is a resubmission of an earlier submission. The following is a list of the peer review reports and author responses from that submission.

Round 1

Reviewer 1 Report

Comments and Suggestions for Authors

Dear Authors,

I did find your images really interesting. However I would advise you to write a paragraph for CBCT and airways in order to connect it with the paragraph you already wrote. This will increase your number of references which will help the overall paper. Please find below a few references:

-Georgiadis T, Angelopoulos C, Papadopoulos MA, Kolokitha OE. Three-Dimensional Cone-Beam Computed Tomography Evaluation of Changes in Naso-Maxillary Complex Associated with Rapid Palatal Expansion. Diagnostics (Basel). 2023 Apr 2;13(7):1322. 

-Tsolakis IA, Kolokitha OE. Comparing Airway Analysis in Two-Time Points after Rapid Palatal Expansion: A CBCT Study. J Clin Med. 2023 Jul 14;12(14):4686. doi: 10.3390/jcm12144686.

-Dastan F, Ghaffari H, Shishvan HH, Zareiyan M, Akhlaghian M, Shahab S. Correlation between the upper airway volume and the hyoid bone position, palatal depth, nasal septum deviation, and concha bullosa in different types of malocclusion: A retrospective cone-beam computed tomography study. Dent Med Probl. 2021 Oct-Dec;58(4):509-514. 

Author Response

Dear

Editor-in-chief /Editorial Board Member/Academic Editor

Diagnostics Journal MDPI

Please find enclosed the revised manuscript entitled “Imaging findings of clinical significance in endodontics during cone beam computed tomography scanning of the upper airway. The A[nterior], B[ilateral], C[shape], D[ual] of mandibular root canals” with ID diagnostics-3808872 that we would like to be considered for publication in Diagnostics Journal-MDPI. Please find also a letter explaining, point-by-point, the changes made in response to the critiques/suggestions/recommendations that we received from peer reviewers.

We sincerely thank the reviewers for their time and thorough review of our manuscript, as well as for the important suggestions and recommendations they have provided us. We have made a concerted effort to respond appropriately to each of the suggestions received from the three reviewers who made up the review committee. We firmly believe that the reviewers' comments and suggestions have significantly and comprehensively improved this manuscript. We hope that you and the reviewers will consider this manuscript suitable for publication in Diagnostics, an MDPI journal.

Comments from Reviewer 1:

Comment 1: I did find your images really interesting. However I would advise you to write a paragraph for CBCT and airways in order to connect it with the paragraph you already wrote. This will increase your number of references which will help the overall paper. Please find below a few references:

-Georgiadis T, Angelopoulos C, Papadopoulos MA, Kolokitha OE. Three-Dimensional Cone-Beam Computed Tomography Evaluation of Changes in Naso-Maxillary Complex Associated with Rapid Palatal Expansion. Diagnostics (Basel). 2023 Apr 2;13(7):1322.

-Tsolakis IA, Kolokitha OE. Comparing Airway Analysis in Two-Time Points after Rapid Palatal Expansion: A CBCT Study. J Clin Med. 2023 Jul 14;12(14):4686. doi: 10.3390/jcm12144686.

-Dastan F, Ghaffari H, Shishvan HH, Zareiyan M, Akhlaghian M, Shahab S. Correlation between the upper airway volume and the hyoid bone position, palatal depth, nasal septum deviation, and concha bullosa in different types of malocclusion: A retrospective cone-beam computed tomography study. Dent Med Probl. 2021 Oct-Dec;58(4):509-514.

Response: Thanks for your comment, we agree. We have, accordingly, modified the paragraph with the suggestions and references mentioned, so that the text after the images now reads as follows:

“The implementation of cone beam computed tomography (CBCT) in dentistry in the late 1990s marked a radical change and a significant advance in dental imaging [31,32]. Notably, CBCT equipment includes software that enables clinicians to visualize, reconstruct, and accurately measure structures such as bones, teeth, and upper airways [33]. These images are stored and man-aged in DICOM format, the international standard protocol for the exchange and management of digital medical images [34], facilitating the visualization of large data sets with specialized DICOM viewers [35]. Moreover, CBCT has proven its usefulness in three-dimensional evaluation of the airways, revealing the intricate relationship between dental anatomy and upper airway morphology, and facilitating the incidental discovery of unusual root canal anatomies [36-40]. Large Field of View CBCT scans enable the exploration and identification of structures in a single exposure, correlating different fields of medicine using the same imaging study for diagnostic, therapeutic, or educational purposes [41]. In contrast, small field of view CBCT is recommended for complex endodontic cases, reducing dispersion and improving image quality in confined spaces like root canals [42,43].”

Reviewer 2 Report

Comments and Suggestions for Authors

Thank you for submitting this case. To meet the Q1 bar, please address the following major points:

1) Provide a succinct but quantitative overview of prevalence for each variant you report (e.g., Vertucci Type III in mandibular anterior teeth; C-shaped mandibular second molars), including bilaterality rates and, where available, population/ethnicity stratification.
Add a summary table with citations (columns: tooth, classification system, prevalence %, bilaterality %, population, source). This will show why the combination observed here is genuinely rare.
2) Re-export figures at high resolution with scale bars, voxel size/slice thickness, and consistent WL/WW. For didactics, add 3D volume rendering and/or segmentation of the canal system (MIP/VR or segmented meshes), plus a coronal–axial–sagittal gallery at standardized increments to illustrate classifications along root levels.
3) Restructure to follow CARE guidelines.
4) Add a concise “Clinical relevance” bullet points on how these findings should change endodontic planning (access shape, scouting sequence, isthmus management, calcium hydroxide if needed, obturation technique), and the risk of missed anatomy if CBCT is not considered.
5) Make the A–B–C–D mnemonic explicit at first mention (A = Anterior, B = Bilateral, C = C-shaped, D = Dual canals) and mirror this labeling in figure captions.
Tighten the Abstract to be case-centric (patient, key findings, clinical implications). Minimize airway epidemiology unless it directly informs endodontic management.
Standardize classification language (e.g., Fan’s C1/C2 by level), tooth notation, and caption style.

Author Response

Dear

Editor-in-chief /Editorial Board Member/Academic Editor

Diagnostics Journal MDPI

Please find enclosed the revised manuscript entitled “Imaging findings of clinical significance in endodontics during cone beam computed tomography scanning of the upper airway. The A[nterior], B[ilateral], C[shape], D[ual] of mandibular root canals” with ID diagnostics-3808872 that we would like to be considered for publication in Diagnostics Journal-MDPI. Please find also a letter explaining, point-by-point, the changes made in response to the critiques/suggestions/recommendations that we received from peer reviewers.

We sincerely thank the reviewers for their time and thorough review of our manuscript, as well as for the important suggestions and recommendations they have provided us. We have made a concerted effort to respond appropriately to each of the suggestions received from the three reviewers who made up the review committee. We firmly believe that the reviewers' comments and suggestions have significantly and comprehensively improved this manuscript. We hope that you and the reviewers will consider this manuscript suitable for publication in Diagnostics, an MDPI journal.

Comments from Reviewer 2:

Comment 1: Provide a succinct but quantitative overview of prevalence for each variant you report (e.g., Vertucci Type III in mandibular anterior teeth; C-shaped mandibular second molars), including bilaterality rates and, where available, population/ethnicity stratification.
Add a summary table with citations (columns: tooth, classification system, prevalence %, bilaterality %, population, source). This will show why the combination observed here is genuinely rare”.

Response: Thank you for pointing this out. We agree with this suggestion.

A concise overview of the prevalence of each reported variant has been added, along with a table containing the required information, thereby greatly enriching the content of the manuscript.

Comment 2: Re-export figures at high resolution with scale bars, voxel size/slice thickness, and consistent WL/WW. For didactics, add 3D volume rendering and/or segmentation of the canal system (MIP/VR or segmented meshes), plus a coronal–axial–sagittal gallery at standardized increments to illustrate classifications along root levels.

Response: Thank you for this suggestion. We agree with this recommendation.

New images with scale bars, voxel size/slice thickness, and WL/WW have been used. Three-dimensional volume renderings of the teeth of interest were added, along with the tomographic sequence with coronal-axial-sagittal galleries and increment lines, which, due to the nature of the manuscript's objective, are difficult to standardize for each tooth, but adequately illustrate the characteristics of the canals at different root levels.  To achieve this, two DICOM viewers different from the one initially used were implemented, which are described and indicated appropriately in the document. Following the modification of the images, the number of figures has been incremented. This allows for a clearer and more detailed visualization of the complex root canal systems, facilitating a better understanding of the anatomical variations and their clinical relevance. With this modification, the manuscript now fulfills its aim of showcasing interesting and educational images, in line with MDPI Diagnostics' guidelines, underscoring the variability of internal root anatomies and the importance of attentive examination of CBCT images, even in non-endodontic contexts, to unlock their full diagnostic potential.

Comment 3: Restructure to follow CARE guidelines.

Response: Thank you for this suggestion. We would like to take this opportunity to point out that the main objective of this manuscript is to describe interesting images, and it is not considered a case report in the strict sense, firstly because no rare disease was identified, and secondly because no medical intervention was performed. However, we agree with the suggestion, as it provides a description of some characteristics of medical interest in the field of endodontics with a possible influence on the clinical endodontic management of the patient (if necessary), for scientific and/or educational purposes, and is now considered as a brief case report of images.

Reference:

  • Nissen, T., & Wynn, R. (2014). The clinical case report: a review of its merits and limitations. BMC research notes, 7, 264. https://doi.org/10.1186/1756-0500-7-264

The CARE 2013 and CARE 2017 guidelines were followed, and we therefore consider that the presentation of these interesting images results in a report with greater scientific support.

The abstract now reads as follow (without the bold sections):

 “3a Introduction: Cone beam computed tomography (CBCT) is a valuable diagnostic tool for evaluating the upper airway and maxillofacial region, providing detailed visualization of airway dimensions, areas of constriction, nasal cavity morphology, and bony and dental structures. This imaging modality is particularly useful in detecting anatomical variations that may impact treatment planning. We present a brief case report highlighting the clinical value of CBCT in identifying significant anatomical variations in endodontics. The incidental detection of complex root canal anatomy on CBCT scans performed for non-endodontic purposes underscores the importance of meticulous image evaluation, further informing the existing literature.

3b Main symptoms and/or important clinical findings: A 23-year-old female patient underwent a CBCT study at the Faculty of Dentistry-UJED, as requested by her general physician, to evaluate her upper airway. Following CBCT imaging evaluation as per standard protocol, an unusual anatomy of some mandibular root canals was revealed, characterized by Vertucci's type III root canals throughout the anterior sextant and bilateral C-shaped mandibular second molars (type C2 according to Fan's classification).

3c The main diagnoses, therapeutic interventions, and outcomes: Based on these findings, a complex root canal anatomy was established, involving the anterior and posterior mandibular teeth. Due to the patient's asymptomatic status, no therapeutic interventions were initiated. This case highlights the importance of recognizing clinically significant anatomical variations, even in the absence of symptoms.

3d Conclusion Given the inherent variability of root and canal anatomy, clinicians should maintain a high index of suspicion for anatomical complexities when planning endodontic therapy. CBCT imaging is a valuable diagnostic tool for identifying these variations, even in non-endodontic scans, thereby facilitating improved patient care and treatment outcomes. Furthermore, documenting and sharing unusual cases through brief case reports can facilitate the detection of similar cases, sensitizing readers to rare or previously unreported findings and contributing to the advancement of medical knowledge.

Following the CARE 2013 and 2017 guidelines, the title has also been modified.

References:

  • Gagnier, J. J., Kienle, G., Altman, D. G., Moher, D., Sox, H., Riley, D., & CARE Group* (2013). The CARE Guidelines: Consensus-based Clinical Case Reporting Guideline Development. Global advances in health and medicine, 2(5), 38–43. https://doi.org/10.7453/gahmj.2013.008
  • Riley, D. S., Barber, M. S., Kienle, G. S., Aronson, J. K., von Schoen-Angerer, T., Tugwell, P., Kiene, H., Helfand, M., Altman, D. G., Sox, H., Werthmann, P. G., Moher, D., Rison, R. A., Shamseer, L., Koch, C. A., Sun, G. H., Hanaway, P., Sudak, N. L., Kaszkin-Bettag, M., Carpenter, J. E., … Gagnier, J. J. (2017). CARE guidelines for case reports: explanation and elaboration document. Journal of clinical epidemiology, 89, 218–235. https://doi.org/10.1016/j.jclinepi.2017.04.026

Comment 4: Add a concise “Clinical relevance” bullet points on how these findings should change endodontic planning (access shape, scouting sequence, isthmus management, calcium hydroxide if needed, obturation technique), and the risk of missed anatomy if CBCT is not considered.

Response: Thank you for this suggestion. We agree with this recommendation.

Thank you for this suggestion. We agree with this recommendation. A section entitled “Clinical relevance” has been added at the end of the manuscript, listing a series of concise bullet points on the clinical relevance of our findings and the implementation of CBCT.

Comment 5: Make the A–B–C–D mnemonic explicit at first mention (A = Anterior, B = Bilateral, C = C-shaped, D = Dual canals) and mirror this labeling in figure captions.

Tighten the Abstract to be case-centric (patient, key findings, clinical implications). Minimize airway epidemiology unless it directly informs endodontic management.

Standardize classification language (e.g., Fan’s C1/C2 by level), tooth notation, and caption style.

Response: Thank you for this suggestion. We agree with this recommendation.

The initials A-B-C-D have been adequately described in the document.

On the other hand, following the recommendation, the abstract was modified to improve the writing following CARE Guidelines.

Likewise, the classification language for anatomical root and canal variants of teeth of interest was standardized, following the classifications provided in scientific literature.

Reviewer 3 Report

Comments and Suggestions for Authors

The authors have observed some unusual root canal types in a 23-year-old female's CBCT and stated that clinicians should always be aware of the possibility of variations in root and canal anatomy when reading CBCT images. The conclusion is correct; however, it lacks novelty and scientific soundness since it was common practice to find some unusual root canal image in patients CBCT scan

Author Response

Dear

Editor-in-chief /Editorial Board Member/Academic Editor

Diagnostics Journal MDPI

Please find enclosed the revised manuscript entitled “Imaging findings of clinical significance in endodontics during cone beam computed tomography scanning of the upper airway. The A[nterior], B[ilateral], C[shape], D[ual] of mandibular root canals” with ID diagnostics-3808872 that we would like to be considered for publication in Diagnostics Journal-MDPI. Please find also a letter explaining, point-by-point, the changes made in response to the critiques/suggestions/recommendations that we received from peer reviewers.

We sincerely thank the reviewers for their time and thorough review of our manuscript, as well as for the important suggestions and recommendations they have provided us. We have made a concerted effort to respond appropriately to each of the suggestions received from the three reviewers who made up the review committee. We firmly believe that the reviewers' comments and suggestions have significantly and comprehensively improved this manuscript. We hope that you and the reviewers will consider this manuscript suitable for publication in Diagnostics, an MDPI journal.

Comments from Reviewer 3:

Comment 1: “The authors have observed some unusual root canal types in a 23-year-old female's CBCT and stated that clinicians should always be aware of the possibility of variations in root and canal anatomy when reading CBCT images. The conclusion is correct; however, it lacks novelty and scientific soundness since it was common practice to find some unusual root canal image in patients CBCT scan”.

Response: Thanks for your comment. You have raised an important point here.

However, in this regard, we would like to mention that it is not common practice to find unusual images of the root canal in patients' CBCT scans, as this requires specific academic and clinical training in the identification of internal and external root anatomy.

This is also because, firstly, CBCT scans are not used exclusively in dentistry (they are not mandatory in any of its specialties), and secondly, because they are not routine examinations in the fields of medicine and dentistry.

On the other hand, the American Association of Endodontists (AAE) and the European Society of Endodontology (ESE) provide guidelines for using Cone Beam Computed Tomography in endodontics, recommending it as a tool for complex cases when 2D imaging is insufficient for diagnosis, particularly for non-healing root canals, trauma, or suspected extra-anatomical root canals, rather than for routine use or screening.

References:

-Patel, Shanon, et al. "European Society of Endodontology position statement: the use of CBCT in endodontics." International endodontic journal 47.6 (2014): 502-504.

- Chugal, N.; Assad, H.; Markovic, D.; Mallya, S.M. Applying the American Association of Endodontists and American Academy of Oral and Maxillofacial Radiology guidelines for cone-beam computed tomography prescription: Impact on endodontic clinical decisions. J Am Dent Assoc. 2024 Jan;155(1):48-58. doi: 10.1016/j.adaj.2023.09.007. Epub 2023 Oct 31. PMID: 37906247.

-Bhatt, M., et al. "Clinical decision‐making and importance of the AAE/AAOMR position statement for CBCT examination in endodontic cases." International endodontic journal 54.1 (2021): 26-37.

In this regard, the novelty is justified because, to our knowledge, this is the first report presenting a mandibular anterior sextant with double root canals, together with the bilateral presence of C-shaped permanent mandibular second molars, which together constitute an interesting and novel description. On the other hand, scientific soundness is achieved with the references cited, which form the basis of the scientific evidence, allowing the findings of this report to be described and compared with the information available worldwide.

This is relevant if we remember that this report specifies that the findings regarding internal root anatomy were thanks to the use of CBCT for exploratory purposes of the upper airway, and not directly as a study aimed at dental or endodontic diagnosis.

To complement this report, a table is included showing the prevalence of radicular canals in the teeth of interest in different populations, and the images have been improved by adding complete tomographic sections and some 2D and 3D representations of the teeth of interest. The changes are highlighted in yellow.